# Association between Improvement of Oral Health, Swallowing Function, and Nutritional Intake Method in Acute Stroke Patients

**DOI:** 10.3390/ijerph182111379

**Published:** 2021-10-29

**Authors:** Michiyo Aoyagi, Junichi Furuya, Chiaki Matsubara, Kanako Yoshimi, Ayako Nakane, Kazuharu Nakagawa, Motoki Inaji, Yuji Sato, Haruka Tohara, Shunsuke Minakuchi, Taketoshi Maehara

**Affiliations:** 1Department of Dysphagia Rehabilitation, Graduate School of Medical and Dental Sciences, Tokyo Medical and Dental University (TMDU), Tokyo 113-8549, Japan; sacra333march@gmail.com (M.A.); k.yoshimi.gerd@tmd.ac.jp (K.Y.); a.nakane.swal@tmd.ac.jp (A.N.); k.nakagawa.swal@tmd.ac.jp (K.N.); h.tohara.swal@tmd.ac.jp (H.T.); 2Department of Geriatric Dentistry, School of Dentistry, Showa University, Tokyo 145-8515, Japan; sato-@dent.showa-u.ac.jp; 3Department of Gerodontology and Oral Rehabilitation, Graduate School of Medical and Dental Sciences, Tokyo Medical and Dental University (TMDU), Tokyo 113-8510, Japan; m.chiakingyo@gmail.com (C.M.); s.minakuchi.gerd@tmd.ac.jp (S.M.); 4Department of Dental Hygiene, Junior College, University of Shizuoka, Shizuoka 422-8021, Japan; 5Department of Neurosurgery, Graduate School of Medical and Dental Sciences, Tokyo Medical and Dental University (TMDU), Tokyo 113-8519, Japan; inaji.nsrg@tmd.ac.jp (M.I.); maehara.nsrg@tmd.ac.jp (T.M.)

**Keywords:** acute stroke, nutrition, oral health, oral function, dysphagia, swallowing

## Abstract

Stroke and poor oral health are common in older people, and the brain injuries associated with stroke are often accompanied by a decline in oral function. In this study, we investigated the characteristics of stroke patients who could not recover oral ingestion until discharge and the association between improved oral health, swallowing function, and nutritional intake methods in acute care. The subjects were 216 consecutive stroke patients who were admitted to Tokyo Medical and Dental University hospital and received oral health management. Nutritional intake, dysphagia, and oral health were evaluated using the Functional Oral Intake Scale (FOIS), Dysphagia Severity Scale (DSS), and Oral Health Assessment Tool (OHAT), respectively. Patients in the tube feeding group (FOIS level 1–2, N = 68) tended to have a worse general condition, fewer functional teeth, and a worse DSS level than those in the oral nutrition group (FOIS level 3–7, N = 148). Multiple analysis with improvement in FOIS score as the dependent variable showed that number of functional teeth (odds ratio [OR]: 1.08, *p* = 0.04) and improved DSS (OR: 7.44, *p* < 0.001) and OHAT values (OR: 1.23, *p* = 0.048) were associated with improvement in nutritional intake methods in acute care. Therefore, recovery of swallowing function and oral health might be important for stroke patients to recover oral ingestion in acute care.

## 1. Introduction

Stroke is a very common disease among older people and is the fourth leading cause of death and the second leading cause for the need of nursing care among Japanese people. In stroke patients, oral function, including swallowing function, is often decreased [1], and also often associated with a decline in brain function due to stroke. Deterioration of oral health and dysphagia can affect the development of aspiration pneumonia, the establishment of oral intake, and the patient’s quality of life (QoL); therefore, appropriate treatment of oral health from the acute phase is required in patients with stroke.

Acute stroke patients are prone to phlegm and retention, xerostomia, and poor oral hygiene due to decreased oral function and disuse [2]. Deteriorated oral health can be improved by early and appropriate oral health management by dental professionals [3]. Furthermore, it has been suggested that active cooperation with dental professionals may help prevent aspiration pneumonia [4]. In recent years, the oral cavity of older people has become more complex due to the use of various prosthetic devices [5], and specialized oral health management is often necessary.

Aspiration pneumonia and respiratory-related diseases are the second most common complication after urinary tract infection in patients with acute stroke [6] and can have a negative impact on subsequent outcomes [7]. The pathogenesis of aspiration pneumonia is related to aspiration of saliva with poor oral hygiene, aspiration of food residue due to poor swallowing function and coughing, and compromised immunity [8]. In the acute phase of stroke, dysphagia is almost inevitable [9,10], even in the case of tube feeding patients due to the subclinical aspiration of unclean saliva containing oral bacteria during sleep, and oral health management from the early stage is important [3].

Nutritional status is also an important issue in patients with acute stroke. It has been reported that 8.2–49% of stroke patients have low nutritional status [11,12]. Poor oral hygiene is strongly associated with malnutrition [13,14]. Furthermore, oral intake using compensatory strategies for swallowing function might help to improve prognosis and prevent of aspiration pneumonia [15,16].

Thus, early stage dysphagia rehabilitation and oral health management are important when recovering oral intake from stroke patients in acute care. In fact, many stroke patients with dysphagia achieve oral intake after 6 months, but dysphagia rehabilitation is essential to recover swallowing function [17]. Furthermore, to facilitate dysphagia rehabilitation, improved oral health is also considered necessary to increase oral intake in patients with acute stroke. Furuya et al. reported that better oral health might be helpful for better oral ingestion level in patients with dysphagia [18]. Although there have been reports on the effects of oral health management on the oral health status in patients with acute stroke [19], the impact on the improvement of nutritional intake has not been elucidated.

It is hypothesized that acute stroke patients who do not acquire oral intake at discharge may present with oral complications as well as worsening stroke severity. Herein, we investigated the characteristics of stroke patients who could not recover oral ingestion until discharge, the association between the improvement of oral health and of swallowing function, and the improvement of the nutritional intake methods in acute care.

## 2. Materials and Methods

### 2.1. Research Participants

Between 1 April 2016 and 31 March 2019, 216 consecutive stroke patients were admitted to Tokyo Medical and Dental University Hospital with cerebral hemorrhage, cerebral infarction, or subarachnoid hemorrhage not caused by trauma, and received multidisciplinary oral health management [19] (Figure 1). Patients with incomplete data were excluded. Data at admission (at the time of the first oral examination within 2–3 days after admission) and at discharge (at the time of the last oral examination within 6 days before discharge) were extracted from the medical records. This study was approved by the Ethics Review Committee of the School of Dentistry of the Tokyo Medical and Dental University (No. D2015-503). Study participants were guaranteed the opportunity to refuse to participate in the study. Data obtained from medical records were consolidated and anonymized for analysis.

### 2.2. Basic Patient Information

Data regarding age, sex, primary disease, duration of hospitalization, number of oral health management interventions by dental professionals, presence or absence of stroke-related surgical procedures, and presence or absence of aspiration pneumonia were collected from medical records, as well as the number of present teeth and functional teeth in the oral cavity. Five dentists and three dental hygienists conducted the oral evaluation following sufficient prior training and calibration. For primary diseases, the presence or absence of cerebral hemorrhage, cerebral infarction, or subarachnoid hemorrhage not caused by trauma was investigated. Daily oral health care was provided by nurses and oral health management by dental professionals, which means professional oral health care by dental hygienists and dental treatment by dentists were performed once or more per week depending on the weekly oral examination described in our previous study [19]. Furthermore, a multidisciplinary conference was regularly held once a week to share information and exchange opinions on oral health. The diagnosis of aspiration pneumonia was made by physicians according to the Japanese Respiratory Society Guidelines for Adult Pneumonia 2017 [20]. The number of present teeth was defined as the number of teeth including residual roots, and the number of functional teeth was defined as the sum of the number of teeth not including residual roots and the number of dentures and other prosthetic teeth to represent mastication ability.

### 2.3. Evaluation Elements in the First Assessment and at Discharge

The evaluation elements included the degree of consciousness impairment, motor function, nutritional intake method, severity of dysphagia, oral health, and biochemical laboratory values (serum albumin: Alb and C-reactive protein: CRP). The Glasgow Coma Scale (GCS) was used to assess the degree of disturbance of consciousness [21]. Motor function was evaluated using the modified Rankin Scale (mRS) [22], and nutritional intake was evaluated using the Functional Oral Intake Scale (FOIS) (Table 1) [23]. The FOIS is a 7-point ordinal scale, where fasting is graded 1 and regular eating is graded 7; it is an international scale for determining the status of eating patterns and nutritional intake. Based on the patient’s general condition and oral function including the severity of dysphagia and oral health status, the physician made the final decision on the type of oral intake by consensus with dentists. The Dysphagia Severity Scale (DSS) [24] was used to assess the severity of dysphagia with swallowing endoscopy. The DSS is a 7-point scale that does not necessarily require swallowing angiography or swallowing endoscopy; lower scores indicate more severe dysphagia [25]; the lower the score, the more severe the dysphagia. Oral health status was assessed using the oral health assessment tool (OHAT) [26], which is a comprehensive oral health assessment tool consisting of eight items: lips, tongue, gingiva/mucosa, saliva, remaining teeth, denture, oral cleanliness, and dental pain, and evaluated at three levels: healthy (0 points), slightly poor (1 point), and diseased (2 points) according to predefined standards. Denture items were evaluated as sound (0 points) if the denture was not necessary from the prosthodontic point of view; the OHAT total score was the sum of each item. Each item was assessed by five dentists and three dental hygienists after adequate prior training at weekly intervals from the initial visit, and data from the evaluation closest to discharge were used for the discharge evaluation.

### 2.4. Classification According to Nutritional Intake at the Time of Hospital Discharge

Study participants were classified into two groups for comparison of their FOIS levels at the time of hospital discharge. The group using tube feeding as the main method of nutritional intake (FOIS level 1–2) was classified as the tube feeding group, and the group mainly taking oral nutrition regardless of diet form (FOIS level 3–7) was classified as the oral nutrition group.

### 2.5. Factors That Influence the Improvement of Nutritional Intake Methods

The presence or absence of improvement in FOIS was used as the dependent variable, clinical significance and multicollinearity were considered, and independent variables were selected. Multiple analysis was performed to examine factors associating FOIS improvement in nutritional intake methods. The group with FOIS improvement was defined as having more than one-point improvement in FOIS at discharge compared to FOIS at admission, and the group without FOIS improvement was defined as having a decrease or maintenance of the score. The degree of improvement in GCS, DSS and OHAT from admission to discharge was scored and entered as an independent variable. For GCS and DSS, the level of improvement at admission was subtracted from the level at discharge, and the scores were described as the level of improvement at GCS and the level of improvement at DSS, respectively. For OHAT, the number of points obtained by changing the positive/negative sign after subtracting the level at discharge from that at admission was used as the OHAT improvement level. A plus (positive) score means the improvement of oral health from admission to discharge.

### 2.6. Statistical Methods

The Mann-Whitney’s U test and χ^2^ test were used for univariate analysis and binomial logistic regression analysis was used for multiple analysis. SPSS Statistics 21.0 (IBM Corp., Armonk, NY, USA) was used as statistical software and all significance levels were set at *p* < 0.05.

## 3. Results

### 3.1. Basic Participants Characteristics

The FOIS score at discharge was used to compare the tube feeding group and the oral feeding group. The number of participants in the oral nutrition group was about twice as high as that in the tube-feeding group. The participant characteristics at admission are shown in Table 2. Significant differences were found between the tube feeding group and the oral feeding group in age (*p* = 0.008), presence of aspiration pneumonia (*p* < 0.001), duration of hospitalization (*p* < 0.001), number of oral health management (*p* < 0.001), and number of functional teeth (*p* < 0.001). There were also significant differences in the GCS (*p* < 0.001), mRS (*p* < 0.001), and DSS (*p* < 0.001) scores and Alb (*p* < 0.001), and CRP (*p* < 0.001) levels.

### 3.2. Comparison of the Tube Feeding Group and the Oral Feeding Group at Discharge

At discharge, there were significant differences in GCS (*p* = 0.003), mRS (*p* < 0.001), DSS (*p* < 0.001) and OHAT total scores (*p* < 0.001) and in Alb (*p* < 0.001) and CRP levels (*p* = 0.001) (Table 3).

### 3.3. Factors Associated with the Improvement of Nutritional Intake Methods

Factors associated with the presence or absence of improvement in the FOIS score (Table 4) were extracted: number of functional teeth (*p* = 0.040, odds ratio [OR] =1.087), level of DSS improvement (*p* < 0.001, OR = 7.441), and level of OHAT improvement (*p* = 0.048, OR = 1.226). This analysis showed that an improvement in the DSS score was seven times more likely to also improve the FOIS score, and a one score improvement in OHAT was associated with a 1.2-fold improvement in FOIS.

## 4. Discussion

In this study, we investigated the characteristics of stroke patients who did not achieve oral food intake at discharge in acute care to examine oral health management that could be associated with the improvement of nutritional intake methods in acute stroke patients. We found that the number of functional teeth, improvement in oral health and in swallowing function were associated with the improvement of nutritional intake at discharge, even after adjusting for the patient’s general condition and the presence of surgery and aspiration pneumonia.

### 4.1. Comparison of the Characteristics of the Participants According to Different Methods of Nutritional Intake at Discharge

Compared with the oral nutrition group, the tube feeding group showed older age, higher incidence of aspiration pneumonia, longer hospitalization, more frequent oral health management by dental professionals, and fewer functional teeth. In general, the older the patient, the greater the risk of periodontal disease and dental caries, and the greater the number of denture incompatibilities [27]. Therefore, it is thought that oral health after the onset of stroke can easily worsen, and the frequency of oral health management increases in the tube feeding patients with a higher average age. The number of cases of aspiration pneumonia was significantly higher in the tube feeding group than in the oral nutrition group. In previous studies, nasal tube feeding has been cited as a factor that increases the incidence of aspiration pneumonia [28]. Langdon et al. reported that more people who were fed by tube after the onset of cerebrovascular disease developed respiratory infections than those who were fed orally and that feeding by tube was an important cause of the development of respiratory infections [29]. In addition, it has been reported that tube feeding is one of the factors that increases the incidence of aspiration pneumonia because the pharynx tends to become unclean due to the adhesion of sputum and bacteria around the tube, and the cough reflex is suppressed [30]. In terms of hospitalization duration, it has previously been reported that compared to the parenteral nutrition group, patients with early oral intake had a shorter treatment period, improved swallowing function, and a lower mortality rate [31]. The results of this study support those of previous reports.

### 4.2. Comparison of Systemic and Oral Items Using Different Nutritional Intake Methods

When comparing systemic and oral items in the tube feeding group and the oral feeding group at discharge, there was no difference in the total OHAT score at admission. In a study of acute hospitalized patients aged 70 years or older, the median OHAT score was 6 points [32]. A previous study of acute stroke patients reported a median OHAT of 4 points at initial diagnosis [19]. The median total OHAT score for both groups in the present study was 4 points at admission as well, showing the same trend as in previous studies, suggesting that oral health was somewhat poor.

Compared to the oral nutrition group at discharge, the tube feeding group tended to have worse general conditions, such as GCS and mRS scores, which supported findings from previous studies indicating that patients with higher stroke severity had greater weight loss due to poor nutritional intake after stroke onset [33]. Furthermore, there was no difference in the total OHAT score between the two groups at admission, but there was a significant difference at discharge.

We hypothesized that the tube-feeding group showed a tendency toward poor oral health. Previous studies have reported that poor oral hygiene is strongly associated with malnutrition [13]. In the present study, the OHAT scores were significantly worse in the tube-feeding group, as hypothesized and in support of previous references.

The DSS score tended to be lower in the tube feeding group. This may be attributed to the nutritional intake method being provided appropriately according to swallowing function. In addition, the tube-feeding group had low Alb levels, suggesting that they were in a low nutritional state. Low nutritional status has been reported to lead to decreased activity and decreased immunity [34], and a decrease in Alb level has been reported to increase dependence on assistance for activities of daily living, prolong hospital stay, and make it difficult to return home [35]. However, since higher CRP levels tend to be associated with lower Alb levels, a more detailed study is needed to determine whether the patients are undernourished, but our results suggest that increased intake of nutrition foods from oral sources may lead to improved nutritional status and influence favorable outcomes.

### 4.3. Factors Influencing the Improvement of Nutritional Intake Methods

Many patients with acute stroke remain on tube feeding at the time of discharge and cannot be transferred to oral nutrition. In the past, approximately half of patients admitted to acute care wards were reported to be undernourished [36]. Of note, modified food for dysphagia, such as paste diets, have lower calories per unit volume than regular diets, and therefore are reported to be insufficient to provide good nutrition [37,38,39]. It is important not only to shift to oral intake but also to shift to a higher level of meal forms. Therefore, to examine the factors that affect the improvement of nutritional intake methods, we performed a multiple analysis in which the independent variables were selected considering clinical significance and multicollinearity, and the dependent variable was the presence or absence of improvement in FOIS.

Our findings revealed that improving swallowing function would be the most important factor in recovering oral ingestion in patients with acute stroke. It has been reported that severe dysphagia can lead to further deterioration of oral function due to poor oral function and the inability to perform oral intake [40]. Dysphagia is often present, especially during the acute phase [9,10]. Our results suggest that the improvement in DSS had a significant impact on the improvement in nutritional intake. Although dysphagia at the time of stroke often resolves spontaneously at a certain level after six months, early intervention in swallowing rehabilitation after the onset of stroke has been reported to improve functional outcome at 6 months and reduce respiratory tract infections [15]. The results of this study also suggested that oral health management aimed at improving swallowing function in the acute phase may be of great significance [17].

In addition, in our study, the number of functional teeth was important for improving the FOIS score. Previous reports on the impact of wearing dentures on swallowing function reported that the improvement of functional teeth by wearing dentures recovers bolus formation and transportation in oral and pharyngeal swallowing [41,42]. Tooth loss has been reported to cause a decrease in masticatory strength and leads to a decrease in nutritional status [43]. Therefore, it was suspected that it might be easier to improve the method of nutritional intake by restoring teeth and maintaining functional teeth with dentures even in the presence of missing teeth.

In addition, improvement level of OHAT was related to the improvement of nutritional intake methods. Poor oral health, which is an endpoint of OHAT, has been reported to be a factor in low nutrition [44], and our findings that good oral health may be essential to recover oral ingestion for acute stroke patients support those of previous studies about oral health of stroke patients. OHAT allows a comprehensive assessment of oral health, suggesting that it might be important to manage not only oral hygiene but also oral function. Furthermore, a study of patients admitted to a recovery ward reported that oral problems were significantly associated with malnutrition [45]. Nutritional status has also been reported to influence the prognosis in the acute phase of stroke [46]. In this way, it is important to improve comprehensive oral health during the acute phase. In the acute phase, the primary goal is to treat the disease, but oral health management in stroke patients is significant for improvement of their quality of life because eating orally would be important for them.

Adjusting for various factors, the current results showed that the presence or absence of surgical procedures or aspiration pneumonia was not associated with the improvement of nutritional intake methods. Nakajoh et al. reported that the incidence of pneumonia was lower with tube feeding than with oral feeding in the case of dysphagia [47], and this was thought to be because the subjects in the present study were mostly patients in the acute phase with reduced swallowing function. From the results of this study, it is clear that the number of functional teeth, swallowing function, and oral health are related to the improvement of nutritional intake methods, and this suggests the importance of improving oral function.

### 4.4. Limitations and Future Research Prospects

There are several limitations to the present study. First, there was a difference in the number of individuals in the tube feeding group and the oral nutrition group in relation to the overall sample size. To not influence the results of the analysis, the objective variable was improving the FOIS score in the multiple analysis. It is necessary to collect additional cases in the tube feeding group in further analyses.

Second, some reports recommend nasal tube feeding as a safe and effective method of feeding patients with dysphagia after acute stroke [48], and the incidence of pneumonia is lower with tube feeding than with oral feeding in patients with dysphagia [47]. In the present study, the oral intake group had the lowest incidence of aspiration pneumonia. However, the results of this study alone do not indicate the recommended method of nutrition with regard to the risk of aspiration pneumonia, and further investigation is needed.

Third, we did not classify patients by lesion location, which cannot be ruled out as a possible influencing factor. Furthermore, there is not enough information about recurrent stroke, other stroke events, and neurological comorbidities in our cohort. Bilateral lesions are associated with more frequent and more severe dysphagia, and brainstem lesions are associated with more frequent aspiration than unilateral hemispheric lesions [49]; therefore, further studies are needed.

Fourth, we used Alb levels from blood test results as an index of nutritional status, but the CRP level also showed high values, so a more detailed examination is necessary to determine whether the patient had a pre-existing low nutritional status based on these results alone. Finally, decreased nutritional status has been reported to lead to decreased immunity, which may affect the length of hospitalization and the development of aspiration pneumonia [34]. Therefore, other nutritional indices such as the body mass index and total protein should be used in future studies.

Fifth, it has been reported that stroke patients often have poor oral function [50], but we have not investigated whether the subjects had received dental care prior to hospitalization. In this study, we consider that the condition of the oral health status before the onset of stroke will have a significant impact, so this is a subject for future research.

## 5. Conclusions

Patients who were unable to receive oral nutrition until discharge from acute hospital care were found to have subsequent poor physical and oral health. Furthermore, it is suggested that oral health management for stroke patients in acute care is important to improve nutritional intake of patients in the acute phase of stroke, because oral health and the number of functional teeth could be related to swallowing function.

## Figures and Tables

**Figure 1 ijerph-18-11379-f001:**
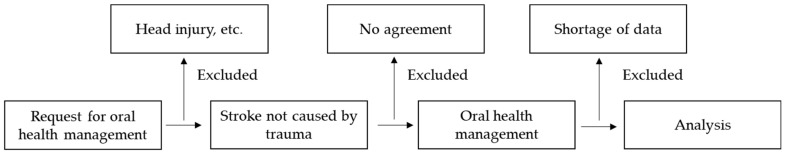
Flow chart of study design.

**Table 1 ijerph-18-11379-t001:** Functional Oral Intake Scale (FOIS).

Level 1	Tube-dependent (including intravenous feeding) and nothing by mouth.
Level 2	Tube-dependent with minimal attempts of food or liquid intake.
Level 3	Tube-dependent with consistent oral intake of food or liquid.
Level 4	Total oral diet of a single consistency.
Level 5	Total oral diet with multiple consistencies, but requiring special preparation or compensation.
Level 6	Total oral diet with multiple consistencies without special preparation, but with specific food limitations.
Level 7	Total oral diet with no restrictions.

**Table 2 ijerph-18-11379-t002:** Comparison between tube feeding group and oral feeding group at the first assessment.

		Tube-Feeding Group (N = 68)						Oral Nutrition Group (N = 148)							
		Mean ± SD	25%	50%	75%	N	%	Mean ± SD	25%	50%	75%	N	%	*p*	Test
Age,	years	67.7 ± 13.4	60.0	70.0	78.8	68		61.9 ± 15.5	49.0	63.0	74.0	148		0.008 **	a
Sex,															
Male,	n					43	63.2					88	59.5	0.598	b
Female,	n					25	36.8					60	40.5		
Primary disease,															
Brain infarct,	n					20	29.4					50	33.8	0.524	b
Brain hemorrhage,	n					28	41.2					50	33.8	0.293	b
Subarachnoid hemorrhage,	n					21	30.9					50	33.8	0.673	b
Coexisting disease,	%					56	82.4					115	77.7	0.434	b
Stroke-related surgical procedures,	%					51	75.0					99	66.9	0.230	b
Aspiration pneumonia,	%					34	50.0					22	14.9	<0.001 **	b
Duration of hospitalization,	days	56.0 ± 33.0	30.5	54.0	72.5	68		35.6 ± 19.1	22.0	32.0	46.8	148		<0.001 **	a
Number of oral health management,	count	10.8 ± 7.9	4.0	9.5	16.8	68		5.3 ± 3.6	2.0	4.0	7.0	148		<0.001 **	a
Number of present teeth,	teeth	20.6 ± 8.7	14.0	24.0	28.0	68		21.2 ± 9.5	16.3	26	28	148		0.268	a
Number of functional teeth,	teeth	20.9 ± 8.9	14.0	24.5	28.0	68		25.2 ± 5.8	25.0	28.0	28.0	148		<0.001 **	a
GCS,	level	5.9 ± 4.2	3.0	3.0	8.0	68		11.57 ± 4.06	9.0	13.0	15.0	148		<0.001 **	a
mRS,	score	4.9 ± 0.4	5.0	5.0	5.0	68		4.24 ± 1.25	4.0	5.0	5.0	148		<0.001 **	a
OHAT total score,	score	4.9 ± 3.1	3.0	4.0	6.8	68		4.5 ± 2.7	2.0	4.0	6.0	148		0.532	a
DSS,	level	1.3 ± 0.7	1.0	1.0	1.0	68		3.1 ± 2.1	1.0	2.0	5.0	148		<0.001 **	a
Alb,	g/dL	3.0 ± 0.6	2.7	3.1	3.4	68		3.4 ± 0.5	3.1	3.4	3.7	148		<0.001 **	a
CRP,	mg/L	5.9 ± 5.2	1.3	4.1	10.0	68		3.1 ± 4.4	0.3	1.3	4.1	148		<0.001 **	a

test a: Mann-Whitney *U* test b: χ^2^ test. ** *p* < 0.01. GCS, Glasgow Coma Scale; mRS, modified Rankin Scale; OHAT, Oral Health Assessment Tool; DSS, Dysphagia Severity Scale, serum albumin; Alb, C-creative protein; CRP.

**Table 3 ijerph-18-11379-t003:** Comparison between the tube feeding group and the oral feeding group at the time of hospital discharge.

		Tube-Feeding Group (N = 68)				Oral Nutrition Group (N = 148)				
		Mean ± SD	25%	50%	75%	Mean ± SD	25%	50%	75%	*p*
GCS,	level	8.8 ± 4.1	5.0	9.0	12.0	14.4 ± 1.3	14.3	15.0	15.0	0.003 **
mRS,	score	4.8 ± 0.5	5.0	5.0	5.0	3.5 ± 1.3	3.0	4.0	4.0	<0.001 **
OHAT total score,	score	3.6 ± 2.6	2.0	3.0	5.0	2.4 ± 1.9	1.0	2.0	4.0	<0.001 **
DSS,	level	1.8 ± 0.8	1.0	2.0	2.0	5.4 ± 1.3	4.3	6.0	7.0	<0.001 **
Alb,	g/dL	3.0 ± 0.5	2.6	3.1	3.3	3.4 ± 0.5	3.1	3.5	3.8	<0.001 **
CRP,	mg/L	3.3 ± 5.6	0.3	1.1	4.5	1.5 ± 2.6	0.1	0.4	1.5	0.001 **

test: Mann-Whitney *U* test. ** *p* < 0.01. GCS, Glasgow Coma Scale; mRS, modified Rankin Scale; OHAT, Oral Health Assessment Tool; DSS, Dysphagia Severity Scale, serum albumin; Alb, C-creative protein; CRP.

**Table 4 ijerph-18-11379-t004:** Logistic regression analysis of improvement of FOIS.

		*p*	Odds Ratio	(95% CI)
Age,	Years	0.866	1.003	0.97–1.04
Sex	0:male, 1:female	0.627	0.788	0.30–2.06
Duration of hospitalization	Days	0.924	0.999	0.97–1.02
Number of teeth present	Teeth	0.076	0.949	0.90–1.01
Number of functional teeth	teeth	0.040 *	1.087	1.00–1.18
Number of oral health management interventions	count	0.783	0.984	0.88–1.11
Improvement level of GCS	score	0.620	0.962	0.82–1.12
Improvement level of DSS	score	<0.001 **	7.441	3.95–14.00
Stroke-related surgical procedures	0:not available, 1:available	0.788	1.146	0.42–3.10
Improvement level of the total OHAT score	score	0.048 *	1.226	1.00–1.50
Aspiration pneumonia	0:not available, 1:available	0.871	0.923	0.35–2.43

Dependent variable: Improvement of FOIS (0: no improvement, 1: improvement). * *p* < 0.05, ** *p* < 0.01. GCS, Glasgow Coma Scale; OHAT, Oral Health Assessment Tool; DSS, Dysphagia Severity Scale.

## Data Availability

The data sets used and analyzed during the current study are available from the corresponding author upon reasonable request.

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
