# Peer review of "Association between Improvement of Oral Health, Swallowing Function, and Nutritional Intake Method in Acute Stroke Patients"

_ijerph, 2021, doi:10.3390/ijerph182111379_

Round 1
Reviewer 1 Report
The manuscript reports the result of a longitudinal study on the predictive factors of oral intake in patients with acute stroke,with a spefic focus on oral status and oral interventions. The study addresses an important topic and is of clinical relevance. However, some points should be addressed to improve the reporting of the study:
- In the title, the asbtract and, then, throughout the paper the authors refers to the effect of oral health management on nutritional intake methods. However, this is not a study on the effects of the treatment as no pre-post assessment is performed, nor a comparison group is available. Therefore, the title and the aim in the whole paper should be revised.
- The dependent variable of the multivariate analysis is sometimes defined as the type of oral intake (oral vs tube feeding) and sometimes as the "improvement" in the type of oral intake (gain in oral intake). This is confusing as they are two different concepts.
- The abstract should be profoundly revised. It is difficult to understand the aim of the study as well as the methods only from reading the abstract.
- Page 2 line 71. The reference to other existing studies on the topic is lacking. Additionally, it should be clearly presented what this paper adds.
- The hypotesis should be added at the end of the introduction
- The title refers to patients with acute stroke, but the word "acute" is lacking in the text of the manuscript. Not even the inclusion criteria specify the acute stage of the patients and the range of days since the onset used to define it.
- Exclusion criteria: did the authors screened for other neurological comorbidities or other stroke events?
- A description of what is defined as dental intervention should be added to the methods
- The criteria used to define the type of oral intake should be specified (e.g. was a clinical or instrumental swallowing assessment performed?)
- Although the overall sample size is quite large, the number of events is 68. As the multivariate analysis included 12 variables, the small sample size of patients with the event should be addressed in the limitations section.
Minor comments:
- a linguistic revision is necessary, some sentences need to be rephrased and grammar errors corrected
- in the abstract the abbreviations DSS and OHAT should be defined
- Page 2 lines 58-63. In this paragraph the authors mix the concept of nutritional status and of type of oral intake. They should be clearly differentiated, thus, the paragraph needs to be revised
- Page 3 line 89, "number of intervention" should be replaced by "number of dental interventions"
- Page 7 line 250-254, please, add references
Reviewer 2 Report
This manuscript is focused on the characteristic of acute stroke patients between tube feeding and oral nutrition, but there was the logical problem of contents, some mistakes and concepts must be clarified and revised. This paper should reconstruct and major revise again.

Reviewer 3 Report
I read with great interest the manuscript entitled: "The relationship between nutritional intake methods and the effect of dental intervention in acute stroke patients".
Attention should be paid to the following points:
1) The authors should indicate in the manuscript and in the abstract the hospital where they have recruited the study participants.
2) Was it analysed whether the patients had previously received dental treatment?
3) Who assessed the oral health of the patients?
Round 2
Reviewer 1 Report
The authors have satisfactory addressed the points raised in the previous review. I would only suggest to revise the abstract in order to provide more details on results, including numbers, which are currently lacking.
Author Response
Thank you for the careful review of our manuscript and for the constructive suggestions. Accordingly, we have made the required changes to the Abstract.
Reviewer 2 Report
The authors have addressed all my questions. In addition, already revised the mistakes and conception, and restructure the contents logically.
Author Response
Thank you for the careful review of our manuscript and the constructive suggestions, which we sincerely appreciate.